# Ubiquitin E3 Ligase AaBre1 Responsible for H2B Monoubiquitination Is Involved in Hyphal Growth, Conidiation and Pathogenicity in *Alternaria alternata*

**DOI:** 10.3390/genes11020229

**Published:** 2020-02-21

**Authors:** Ye Liu, Jingjing Xin, Lina Liu, Aiping Song, Yuan Liao, Zhiyong Guan, Weimin Fang, Fadi Chen

**Affiliations:** State Key Laboratory of Crop Genetics and Germplasm Enhancement, Key Laboratory of Landscaping, Ministry of Agriculture and Rural Affairs, College of Horticulture, Nanjing Agricultural University, Nanjing 210095, China; liuye@njau.edu.cn (Y.L.); 2016204037@njau.edu.cn (J.X.); 2017104100@njau.edu.cn (L.L.); liaoyuan@njau.edu.cn (A.S.); aiping_song@njau.edu.cn (Y.L.); guanzhy@njau.edu.cn (Z.G.); fangwm@njau.edu.cn (W.F.)

**Keywords:** H2B monoubiquitination, AaBre1, H3K4 methylation, transcription regulation, *Alternaria alternata*

## Abstract

Ubiquitination is one of several post-transcriptional modifications of histone 2B (H2B) which affect the chromatin structure and, hence, influence gene transcription. This study focuses on *Alternaria alternata,* a fungal pathogen responsible for leaf spot in many plant species. The experiments show that the product of *AaBRE1,* a gene which encodes H2B monoubiquitination E3 ligase, regulates hyphal growth, conidial formation and pathogenicity. Knockout of *AaBRE1* by the homologous recombination strategy leads to the loss of H2B monoubiquitination (H2Bub1), as well as a remarkable decrease in the enrichment of trimethylated lysine 4 on histone 3 (H3K4me3). RNA sequencing assays elucidated that the transcription of genes encoding certain C2H2 zinc-finger family transcription factors, cell wall-degrading enzymes and chitin-binding proteins was suppressed in the *AaBRE1* knockout cells. GO enrichment analysis showed that these proteins encoded by the set of genes differentially transcribed between the deletion mutant and wild type were enriched in the functional categories “macramolecular complex”, “cellular metabolic process”, etc. A major conclusion was that the *AaBRE1* product, through its effect on histone 2B monoubiquitination and histone 3 lysine 4 trimethylation, makes an important contribution to the fungus’s hyphal growth, conidial formation and pathogenicity.

## 1. Introduction

The posttranslational modification of histones, involving one or more of monoubiquitination, methylation, phosphorylation and acetylation, is a key epigenetic mark, which can turn genes on or off with no change in the DNA sequences [1,2]. Monoubiquitination is related to the alteration of the molecular functions or the subcellular localization of the target proteins, which is different from polyubiquitination, which is closely related to proteasome-mediated degradation [3,4]. The monoubiquitination of histone 2B (H2B) has been documented to govern both the growth and development of a range of eukaryotic organisms [5,6]. In *Arabidopsis*, two E3 ubiquitin ligases are responsible for H2B monoubiquitination [5], while, in yeast, a single E3 ligase is involved [7]. The mediation of H2B monoubiquitination by the protein Bre1 has been noted in both yeast and a range of animal and plant species [7,8,9,10]. H2B monoubiquitination is necessary for the methylation of the fourth lysine residue of histone H3 (H3K4), an additional epigenetic mark exerting a major influence over gene expression [9,11,12]. Bre1-mediated H2B monoubiquitination has been widely studied in animals, plants and yeast cells, but little is known about their functions in plant pathogenic fungi. In recent years, the roles histone methylation played in fungal growth, conidiation, secondary metabolites and pathogenesis have been widely reported in *Aspergillus fumigatus* [13,14], *Magnaporthe grisea* [15], *Fusarium graminearum* [16,17], while the functions of histone monoubiquitination still remain elusive. 

*Alternaria alternata* is a necrotrophic fungus, which is always found in soil and various decaying plant materials [18]. The infection of *A. alternata* can compromise economic yield and/or end-use quality and, more seriously, *A. aternata* has been regarded as the most important mycotoxin-producing species among all *Alternaria spp.* that seriously threatens human health [19,20]. In *Chrysanthemum* (*Chrysanthemum morifolium*), an ornamental species of high economic value, *A. alternata* is the pathogen responsible for black spot disease, which can result in major economic losses to the producers. Despite the huge economic loss caused by *Chrysanthemum* leaf black spot disease, efficient prevention and control strategies of the disease have not been fully developed, and this may be partly due to our limited understanding of *A. alternata* pathogenetics. Efforts to understand the basis of fungal pathogenicity have included the construction of a number of gene knockouts. In recent years, following the development of the molecular biological methods of *A. alternata*, the functions of a few genes in *A. alternata* genome were elucidated by gene knockout. For example, glutathione peroxidase 3 (AaGPx*3*) works on the detoxification of cellular stresses in *A. alternata* [21]; LaeA and VeA played key roles in fungal growth and mycotoxin production [22] and Csn5 is necessary for the conidiation and pathogenesis in the *A. alternata* tangerine pathotype [23]. 

As yet, no genes encoding proteins involved in histone modification have been characterized in *A. alternata*. The present study was intended to explore the influence of H2B monoubiquitination over fungal growth and pathogenicity in *A. alternata*. Through the homologous recombination strategy, the E3 ligase Bre1 homolog AaBre1 was knocked out in *A. alternata*. Furthermore, we found that AaBre1 is responsible for H2B monoubiquitination and involved in fungal growth, conidiation and pathogenicity. In order to further elucidate the role AaBre1 plays in biological and pathogenesis processes, RNA sequencing analyses were performed between the wild type (WT) and *AaBRE1* deletion mutant ΔAaBre1, which provided useful information in gene transcription regulated by H2B monoubiquitination.

## 2. Materials and Methods

### 2.1. Fungal Strains and Medium

A wild type (WT) strain of *A. alternata* was isolated from *Chrysanthemum* leaves exhibiting the black spot disease. Both WT and its derived strains in this study were cultured on potato dextrose agar (PDA), containing 200 g/L potato, 20 g/L glucose and 20 g/L agar, which were used for mycelial growth and sporulation analysis. For long-term storage, mycelial cultures were immersed in 15% v/v glycerol and held in a refrigerator at −80 ℃.

### 2.2. Generation of AaBre1 Deletion Mutants

An AaBre1 deletion fragment was generated by using the double-joined PCR method [24]. For *AaBRE1*, the primer pairs AaBre1-UP-F/-R and AaBre1-DOWN-F/-R (Appendix A), respectively, were used for amplifying the upstream fragments (1.03 kbp) and downstream fragments (1.09 kbp). The hygromycin phosphotransferase fragment (*HPH*) was amplified using the primer pairs HPH-F/-R (Appendix A). The resulting PCR products were connected by using the double-joined PCR method. The fragments used for gene knocking out were amplified with the primer pairs AaBre1-KO-F/-R (Appendix A) and the resulting amplicon were transformed into protoplasts prepared from the wild type (WT) cells. The protoplasts were cultured on PDA containing 100 mg/L hygromycin in order to select those carrying the deleted form of AaBre1 (ΔAaBre1), and surviving protoplasts were subjected to both a genomic PCR and a quantitative real-time PCR (qRT-PCR) assay directed at *AaBRE1,* driven by the primer pair AaBre1-ID-F/-R and AaBre1-QRT-F/-R (Appendix A).

### 2.3. AaBre1 Complementation and Microscopic Examinations of AaBre1 Protein Localization

To further confirm that differences in phenotype between ΔAaBre1 and WT were due to the loss of *AaBRE1*, we constructed the complement vector pYF11-*AaBRE1*-C and then transformed into the mutant ΔAaBre1. The complement fragment *AaBRE1*-C was amplified by using the primer pair AaBre1-GFP-F/-R (Appendix A), which were co-transformed with the *Xho* I-digested PYF11 vector containing *GFP* [25] into DH5α by using the ClonExpress^®^ II One-Step Cloning Kit (catalog no. c112, Vazyme, China). In order to guarantee the exactness of the *AaBRE1* sequence, the gene in the plasmid pYF11-*AaBRE1*-C was sequenced. Then, the complement plasmid pYF11-*AaBRE1*-C were inserted into the genome of ΔAaBre1 by protoplast transformation and the selection agent was geneticin. In order to observe the localization of AaBre1, fresh mycelia were harvested and the Zeiss LSM800 ultrahigh resolution confocal microscope (Carl Zeiss AG, Germany) were used for observing the subcellular localization of AaBre1. Meanwhile, 4’6-diamidino-2-phenylindole (DAPI) was used to counter-stain the nuclei.

### 2.4. RNA Extraction and Quantitative Real-Time PCR

Fresh mycelia of each strain were cultured in potato dextrose broth (PDB) containing 200 g/L potato and 20 g/L glucose at 25 °C for twenty-four hours in the dark, and then were washed and harvested to extract total RNA. The assays of total RNA extraction, cDNA synthesis and the qRT-PCR were performed as described previously [26]. TaKaRaRNAiso Reagents were used for cDNA Synthesis and qRT-PCR assays. The primers used for the expression levels of genes are listed in Appendix A. The primer pair AaACT-QRT-F/-R (Appendix A) were used to amplify the *ACTIN* gene, which was used as a reference in each sample. Each assay was represented by three biological replicates.

### 2.5. Western Blot Assays

Western blot assays were implemented to detect H2B monoubiquitination and H3K4 trimethylation in WT and ΔAaBre1 cells. Both fungal strains were cultured in the PDB medium at 25 °C for twenty-four hours. Then, mycelia were washed and harvested for protein extraction. The experiments of protein extraction and western blot assays were conducted as described previously [17,27]. Antibodies anti-histone H3 (catalog no. ab8580; AbCam), anti-H3K4 tri-methyl (catalog no. ab8580; AbCam), and anti-H2Bub1 (catalog no. 39238; Active Motif) were used at a dilution of 1:5000 for western blot assays. Each assay was represented by at least three biological replicates.

### 2.6. Conidiation, Growth and Pathogenicity Assays 

Conidia counts were conducted on both WT and ΔAaBre1 cultures grown on PDA for seven days, in darkness, at 25 °C. Colony morphology was recorded photographically from plates cultured at 25 °C for seven days. The inoculation assays were performed by using the mycelium of each strain, cultured at 25 °C for one day, in a shaker, at 180 rpm. Mycelia were collected from 0.5 mL mycelia suspension and then inoculated onto *Chrysanthemum* leaves. The pathogenicity experiments of WT and ΔAaBre1were carried out on ten *Chrysanthemum* leaves. The leaves of the *Chrysanthemum* were cultured at 25 ± 2 °C under 95%–100% humidity after inoculation. The necrotic lesions on each inoculated *Chrysanthemum* leaf were imaged after seven days’ inoculation. 

### 2.7. RNA Sequencing Analyses 

Six libraries (Aa_1, Aa_2, Aa_3, BR_1, BR_2 and BR_3) were constructed, comprising three replicates of each of WT and ΔAaBre1. The clean reads in which the proportion of non-called reads was > 10% and low-quality (≤ 20) reads represented > 50% of the reads that were used for subsequent analysis. The clean reads were mapped onto the *A. alternata* genome (http://fungi.ensembl.org/Alternaria_alternata/Info/Index) by HISAT2 (https://daehwankimlab.github.io/hisat2/). The frequency of occurrence of individual reads was normalized by conversion to Fragments Per Kilobases per Millionreads (FPKM) [28]. Differential expression gene (DEG) analysis was performed by using DESeq2 software [29]: The p-adjusted *p*-value (padj) provides a criterion to determine the *p*-value threshold in multiple tests and analyses. In the study, DEGs were considered when padj < 0.05 and the absolute value of log 2 induction ratios of the deletion mutants ΔAaBre1 compared with the wild type (WT) was ≥ 1.0. Functional analysis of the DEGs relied on the Pfam (http://pfam.xfam.org), GO (geneontology.org), and KEGG (www.genome.jp/kegg/kegg1.html) databases. The *p*-value for each represented GO term was corrected via the Benjiamini & Hochberg false discovery rate (FDR) correction method. The Pfam database (http://pfam.xfam.org) and National Center for Biotechnology Information (NCBI) conserved domain search (https://www.ncbi.nlm.nih.gov/Structure/cdd/wrpsb.cgi) were used to assign the functional domains of DEGs. The data were analyzed on the free online Majorbio Cloud Platform (www.majorbio.com). The RNA sequencing data has been submitted to the SRA database (https://submit.ncbi.nlm.nih.gov/subs/sra/) and the BioProject accession number is PRJNA597684.

### 2.8. Identification of Transcription Factors and Secreted Protein Effectors

Transcription factors (TF) were identified using HMM-scan software and were annotated using the Blast software. A heatmap of differentially expressed transcription factors was constructed using the R package ‘Pheatmap’ (cran.r-project.org/web/packages/pheatmap/pheatmap.pdf). The prediction of secreted proteins was identified on the basis of the three following features: (a) the N-terminal signal peptide; (b) no transmembrane domains; (c) the protein is secreted to the extracellular level [30,31]. First, the N-terminal signal peptide was predicted by SignalP-4.1 (http://www.cbs.dtu.dk/services/SignalP-3.0/), then using TMHMM-2.0c (http://www.cbs.dtu.dk/services/TMHMM-2.0/) to remove the proteins containing the transmembrane domains. Phobius101 (http://phobius.sbc.su.se/) was also used to further confirm that the secreted protein effectors have an N-terminal signal peptide, but no transmembrane domains. Finally, the subcellular localization of the candidate proteins above were predicted by ProtComp v3 (http://www.softberry.com/berry.phtml?topic=protcompan&group=help&subgroup=proloc) and the secreted proteins predicted to reach the outer membrane were retained.

## 3. Results 

### 3.1. Identification and Knockout of AaBre1 in A. alternata 

A Blast search of the *A. alternata* geonome sequence identified the sequence CC77DRAFT_964290 in the *A. alternata* genome (http://fungi.ensembl.org/Alternaria_alternata) as a probable Bre1 homolog: the sequence was, therefore, denoted as AaBre1. The genome sequence predicts the length of this gene to be 2308 bp. Then, we cloned and sequenced the genomic DNA sequence of AaBre1 in full length in the wild type (WT). There was no change in the amino acid sequence region of AaBre1 from the *A. alternata* strain WT and *A. alternata* genome reported. A phylogenetic analysis indicated AaBre1 to be closely related to the *Aspergillus nidulans* protein AnBre1 (Figure 1A). Based on domain characteristics analyzed by the Pfam and NCBI conserved domain search database, AaBre1 was predicted to include the same set of putative domains as its yeast homolog ScBre1 (Figure 1B).

Then, based on the sequence of *AaBRE1*, we constructed the fragments used for gene deletion (Figure 2). The homologous recombination procedure succeeded in replacing a 959 bp fragment of *AaBRE1* with the sequence encoding hygromycin phosphotransferase fragment (*HPH*), thereby generating the strain ΔAaBre1, an *AaBRE1* loss-of-function deletion mutant (Figure 3A). We performed PCR assays to identify the *AaBRE1* deletion mutants by using the gene-specific primer pair, AaBre1-ID-F/-R (Appendix A), which amplified a 1657 bp fragment from the *AaBRE1* deletion mutants and 1323 bp fragment from the wild type (WT) (Figure 3B). Moreover, a qRT-PCR assay also verified the *AaBRE1* deletion in the mutants △AaBre1 (Figure 3C). 

### 3.2. Subcellular Localization of AaBre1 in A. alternata

The subcellular localization of the protein gives us some clues to its biological functions. In order to explore the subcellular localization of AaBre1 in *A. alternata*, the full length of the *AaBRE1* coding sequence under the native promoter was fused with green fluorescent protein (GFP), a copy of *AaBRE1-GFP* was inserted into the genome of △AaBre1 and we acquired the complemented strain △AaBre1-C (Figure 3B). The restocking of ΔAaBre1 with *AaBRE1* under the native promoter recovered the growth defects of ΔAaBre1 on the PDA medium (Figure 3A). 

The mycelium produced by the ΔAaBre1-C strain featured multiple near-circular green fluorescence (Figure 4), which implies that the protein AaBre1 is nuclear-localized. To further confirm our inference, we performed simultaneous nuclear staining by 4′-6-diamidino-2-phenylinodle (DAPI). On the merged image, the GFP and DAPI staining were overlapped (Figure 4), which indicated that AaBre1 was nuclear-localized in *A. alternata*.

### 3.3. AaBre1 Regulates H2B Monoubiquitination and H3K4 Trimethylation

Western blot assays were used to identify the role AaBre1 plays in both H2B monoubiquitination and H3K4 trimethylation in *A. alternata*. Antibodies that recognize H2B monoubiquitination and H3K4 trimethylation were used in our study. As shown in Figure 5, H2B monoubiquitination was blocked in the ΔAaBre1 mutants. These results indicated that AaBre1 was required for H2B monoubiquitination and positively regulates H2B monoubiquitination in *A. alternata*. As reported, H2B monoubiquitination is necessary for H3K4 methylation [11,12]. We also investigated the levels of H3K4 trimethylation between WT and ΔAaBre1. As shown in Figure 5, the ΔAaBre1 mutant showed drastically decreased H3K4 trimethylation. These results clearly indicate that AaBre1 plays a vital role in both H2B monoubiquitination and H3K4 trimethylation in *A. alternata*.

### 3.4. AaBre1 is Required for Hyphal Growth, Conidiation and Pathogenicity

Mycelium growth of ΔAaBre1 was about 30% less than that of WT when the strains were cultured on PDA (Figure 6A). To elucidate the role AaBre1 plays in asexual development, each strain was inoculated on PDA for seven days. After seven days’ growth, fewer conidia were produced by the deletion mutants ΔAaBre1 than by the wild type (WT) (Figure 6B). The results suggest that AaBre1 is essential for normal mycelial growth and conidiation in *A. alternata*. To analyze the role AaBre1 plays in pathogenicity, infection assays with *Chrysanthemum* leaves were performed. In the pathogenicity assays, ΔAaBre1 showed reduced virulence, and necrotic lesions were limited to the inoculated position after ΔAaBre1 inoculation. However, severe and expanding necrotic lesions were formed in *Chrysanthemum* leaves after WT inoculation (Figure 6C). The results above indicate that AaBre1 is necessary for the full virulence of *A. alternata*. 

### 3.5. RNA Sequencing Analyses Reveal Roles AaBre1 Played in Various Pathways

With the development of epigenetics, increasing studies have reported the relationship between histone modification and gene transcription [32]. H2B monoubiquitination and H3K4 trimethylation play significant roles in the regulation of gene transcription [33]. In order to further explore the roles AaBre1 plays in gene transcription and identify candidate genes regulated by H2B monoubiquitination and H3K4 trimethylation in *A. alternata*, RNA sequencing analyses were performed to compare the transcriptomes of the ΔAaBre1 with WT. These were done in order to identify the range of genes regulated by H2B monoubiquitination and H3K4 trimethylation. Each strain has three replicates. The Q20 and Q30 of the set of clean reads were, respectively, >98.71% and >95.81% (Appendix A). After filtering, over 80% of the clean reads could be mapped onto the *A. alternata* genome (Appendix A), which suggests that the samples are comparable. Transcript abundances based on FPKM were highly correlated, and the correlation between two samples among the three replicate samples from the wild type (WT) cells or the mutants, ΔAaBre1, was >0.99 (Figure 7A). A principal component analysis (PCA) implied that a high level of similarity obtained between the three WT replicates, as well as between the three ΔAaBre1 replicates (Figure 7B). The above results showed that the three replicate samples of each strain were reliable. In the RNA sequencing assay, a total of 1604 DEGs was recognized: 486 genes were down-regulated and 1118 genes were up-regulated in ΔAaBre1 compared to the wild type (WT) (Figure 7C). A GO enrichment analysis suggested that functional categories enriched in differentially expressed genes were “macramolecular complex”, “cellular metabolic process”, and so on (Figure 7D).

### 3.6. AaBre1 Regulates the Expression of Transcription Factors

Because transcription factors play central roles in gene regulation, we further investigated transcription factors in *A. alternata* and 160 transcription factors belonging to 24 families were identified in the *A. alternata* genome (Figure 8A). Of these transcription factors, 17 differentially expressed transcription factors were identified, as shown in Figure 8B,C. Among the differentially expressed transcription factors, nine were down-regulated and eight were up-regulated in the mutant strain ΔAaBre1 (Figure 8B,C). The former set of genes belonged to the gene families NDT80/PhoG, ZBTB and zinc-finger C2H2 (zf-C2H2). Moreover, three genes containing nuclear localization signals and tandemly arranged C2H2 zinc-finger domains were down-regulated in the ΔAaBre1 mutants. The identification of down-regulated zf-C2H2 transcription factor lays the foundation for further studies to reveal the regulation mechanism that H2B monoubiquitination is involved in fungal growth, conidiation or pathogenesis.

### 3.7. AaBre1 Regulates Expression of Fungal Effectors

The colonization of plants by fungi is governed by hundreds of secreted fungal effector molecules [34]. Microbial pathogens transfer effector proteins to inhibit Pathogen Associated Molecular Pattern (PAMP)-triggered host immunity and establish infections [35]. In our study, the effectors were identified in the *A. alternata* genome. Using SignalP4.0, a total of 1431 (10.5%) out of 13658 open reading fragments (ORFs) were predicted to be typically secreted proteins. TMHMM2.0c was used to predict transmembrane domains, which showed that 1186 have no predicted transmembrane domain and 245 have at least one predicted transmembrane helix among the 1431 total predicted secreted proteins. Furthermore, the 1186 genes above were identified as secreted proteins without a transmembrane domain. By using Phobius101, another 358 proteins containing signal peptides without transmembrane terminals were also identified. Finally, all the secreted proteins were further analyzed by Protcomp v9.0, which showed that the subcellular localization of 858 candidate proteins were extracellular. The results above indicate that the 858 proteins (6.28% of the proteome) were identified as possible effectors in *A. alternata* (Figure 9A).

The transcriptomic comparison between the WT and ΔAaBre1 strains showed that 39 of the possible effectors were less strongly transcribed in the mutant ΔAaBre1 (Figure 9B,C). A joint analysis, based on the Pfam database and on the NCBI conserved domain search database, was carried out to assign functional domains to these down-regulated genes. In addition, Glyco_hydro, Pectate_lyase and the chitin-binding domain were identified in five of the predicted products of the 39 DEGs, as shown in Figure 9D. Plant cell wall-degrading enzymes (PCWDEs), containing the Glyco_hydro or Pectate_lyase domain, help most necrotrophs form inconspicuous appressoria and penetrate the plant cuticle [34]. In the study, three differentially expressed candidate PCWDEs (CC77DRAFT_1011316, CC77DRAFT_1061819 and CC77DRAFT_237249) were identified. Effectors which harbor a chitin domain, such as Avr4 [36,37] and Ecp6 [35,38], are known to protect fungal cell walls against hydrolysis by plant chitinases. Among our identified DEGs, two genes (CC77DRAFT_1067253 and CC77DRAFT_1055189) containing the chitin-binding domain are significantly down-regulated, which are also important candidate genes involved in pathogen pathogenesis.

## 4. Discussion

Gene expression is known to be strongly influenced by histone modification in both animal and plant systems [1]. The monoubiquitination of H2B is one of the most important types of post-translation modification. Both *HUB1*/*HUB2* in *Arabidopsis thaliana* and *BRE1* in yeast encoding enzymes have been established to drive H2B monoubiquitination but, as yet, little attention has been paid to equivalent processes in fungi. Here, the focus was to reveal the function of an *A. alternata BRE1* homologous gene, *AaBRE1*. By comparing a WT strain with one lacking the functional copy of *AaBRE1,* it was possible to conclude that the product of *AaBRE1* is important for the fungus’s vegetative growth, its ability to develop conidia and its pathogenicity with respect to *Chrysanthemum*. In the ΔAaBre1 strain, ubiquitinated H2B was not produced (Figure 5A, B), which clearly implies that AaBre1 is required for H2B monoubiquitination, as is also the case for yeast Bre1.

Phylogenetic analysis showed that the ubiquitin E3 ligases and homologues involved in H2B monoubiquitination belong to two main clades. Bre1 in yeast and its homologues in fungi belong to one clade, while HUB1 and HUB2 in plants fall into another. The results indicate that the process of H2B monoubiquitination likely differs at the mechanistic level between plants and microorganisms. In yeast, H2B monoubiquitination appears to be related to H3K4 methylation in active chromatin and H2Bub1 is necessary for H3K4 methylation [7,39], but this does not seem to be the case in plants. In *Arabidopsis thaliana*, for example, the genome-wide level of H3K4 methylation in WT plants is substantially similar to that of mutants unable to ubiquitinate H2B [5]. Meanwhile, in rice, H2B monoubiquitination regulated by OsHUB1 and OsHUB2 was in concert with H3K4 dimethylation [6]. The present experiments have shown that in *A. alternata*, the loss of H2B monoubiquitination resulted in a reduced level of H3K4 trimethylation. These studies indicate that the relationship between H2B moniubiquitination and H3K4 methylation varies among different species.

H2Bub1, which has been identified as a marker of active chromatin, exerts a strong influence over gene expression. The RNA sequencing data acquired here indicated that genes encoding certain transcription factors were significantly down-regulated in strains lacking a functional copy of AaBre1. The loss of the C2H2-type transcription factor has been documented to compromise hyphal growth, conidial formation and virulence in *Phytophthora sojae* [40], *Aspergillus fumigatus* [41], *Verticillium dahlia* [42], *Ustilago maydis* [43], and *Fusarium graminearum* [44]. Three genes identified as likely members of the C2H2 family (since they harbored both tandemly arranged C2H2 zinc-finger domains and a nuclear localization signal) were suppressed in the ΔAaBre1 mutant strain. The identification of the zinc-finger C2H2 transcription factor, which was down-regulated, lays the foundation for further studies to reveal the regulation mechanism that H2B monoubiquitination is involved in fungal growth, conidiation or pathogenesis. In the study, we have also pointed out that AaBre1 could be linked to the expression of fungal effectors, as the deletion of *AaBRE1* lead to a down-regulation of secreted protein effectors. Plant cell wall-degrading enzymes (PCWDEs) and effectors containing the chitin domain play important roles during infection [34,38]. In the present study, we identified three differentially expressed candidate PCWDEs (CC77DRAFT_1011316, CC77DRAFT_1061819 and CC77DRAFT_237249). Among the identified DEGs, two genes (CC77DRAFT_1067253 and CC77DRAFT_1055189) containing the chitin-binding domain were also found, which are also important candidate genes involved in pathogen pathogenesis. It was apparent that the product of *AaBRE1* affected the expression of various PCWDEs and proteins harboring a contain chitin domain. The identification of effector proteins regulated by AaBre1 give a basis for further revealing the pathogenic mechanism of *A. alternata*.

Collectively, a model that reveals the function of AaBre1 was proposed (Figure 10). We identified the E3 ubiquitin ligase AaBre1, which was involved in the monoubiquitination of H2B in *A. alternata*. Moreover, AaBre1 was nuclear-localized and related to the enrichment of H3K4 trimethylation. Our results also indicate that AaBre1 regulates hyphal growth, conidiation and pathogenicity in *A. alternata*, the causal agent of black spot disease in *Chrysanthemums*. In addition, by high-throughput sequencing technology, we identified a series of genes that were down-regulated in ΔAaBre1 that may be regulated by H2B monoubiquitination, including zf-C2H2 family transcription factors, plant cell wall-degrading enzymes and chitin-binding proteins, which may be associated with hyphal growth, conidiation and pathogenicity in *A. alternata*. We speculate that AaBre1 may regulate hyphal growth, conidiation and pathogenicity by influencing the enrichment of H2B monoubiquitination and H3K4 tri-methylation to regulate the expression of transcription factors and effectors. Our study opens distinctive pathways regarding the roles H2Bub1 plays in regulating fungal growth, development and virulence in plant pathogenetic fungi. 

## Figures and Tables

**Figure 1 genes-11-00229-f001:**
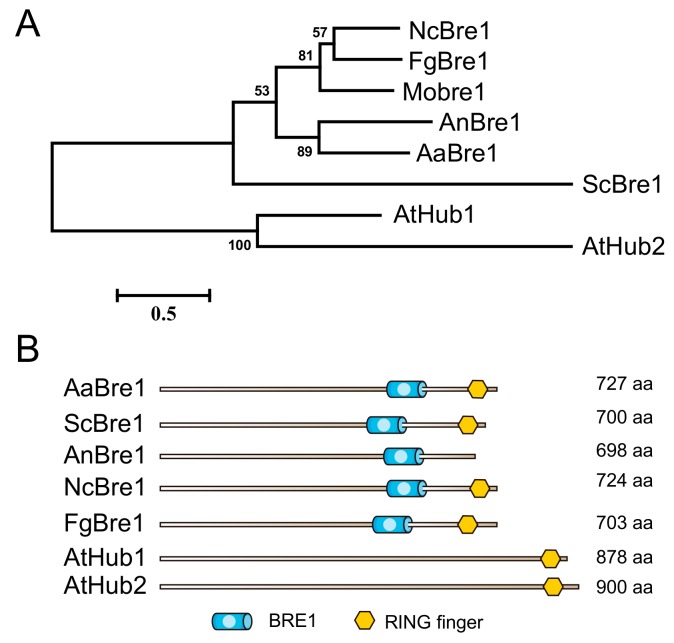
Identification of E3 ligase AaBre1 in *A. alternata*. (**A**) Phylogenetic analysis of Bre1 homologues in several organisms. The alignment was performed with ClustalW2 and the MEGA program, version 5.1, was used for a phylogenetic analysis using the maximum-likehood tree method. *Neurospora crassa* (Nc); *Aspergillus nidulans* (An); *Saccharomyces cerevisiae* (Sc); *Arabidopsis thaliana* (At). (**B**) Domains organization of Bre1 homologues. The domains identified by using Pfam are BRE1 and RING finger.

**Figure 2 genes-11-00229-f002:**
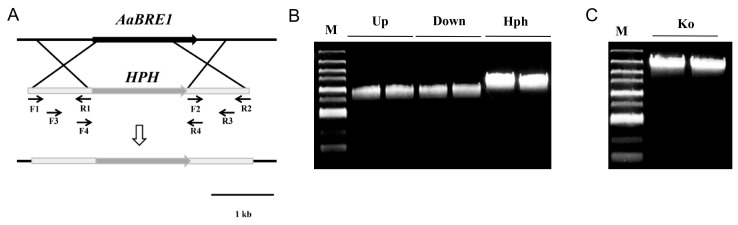
Schematic representation of the *AaBRE1* deletion strategy. (**A**) *AaBRE1* and hygromycin phosphotransferase fragment (*HPH*) are denoted by large black and grey arrows, respectively. Arrows indicate the annealing sites of PCR primers. (**B**) PCR products for the *AaBRE1* gene replacement construct. DNA size marker (M); upstream of *AaBRE1* gene replacement fragment (Up); downstream of *AaBRE1* gene replacement fragment (Down); hygromycin phosphotransferase fragment (HPH). (**C**) PCR products for the full length of *AaBRE1* gene replacement construct (Ko).

**Figure 3 genes-11-00229-f003:**
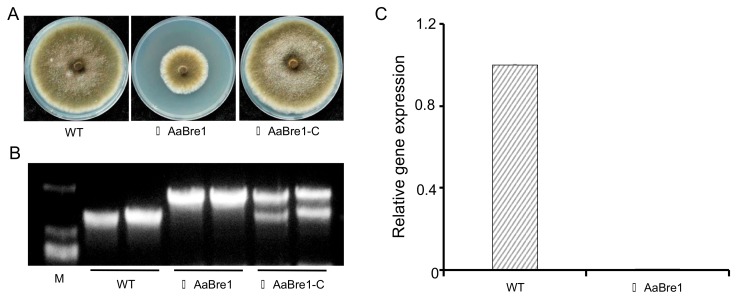
Characterization of ΔAaBre1 and ΔAaBre1-C strains. (**A**) Wild type (WT), ΔAaBre1 and ΔAaBre1-C were grown on PDA medium at 25 ℃ for seven days. (**B**,**C**) Molecular verification of the mutant strains, using (**B**) a gDNA-based PCR assay, (**C**) a quantitative real-time PCR (qRT-PCR) assay.

**Figure 4 genes-11-00229-f004:**
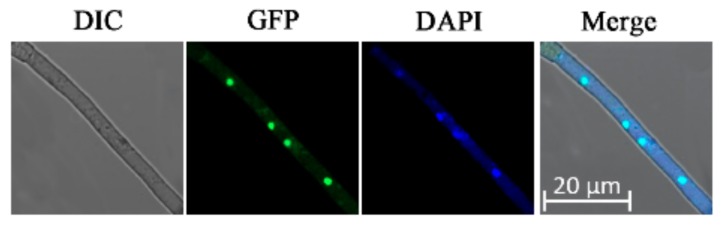
Subcellular localization of AaBre1 in *A. alternata*. AaBre1 was mainly localized to the nucleus. 4’6-diamidino-2-phenylindole (DAPI) was used to observe nuclei. Differential interference contrast (DIC); bars, 20 μm.

**Figure 5 genes-11-00229-f005:**
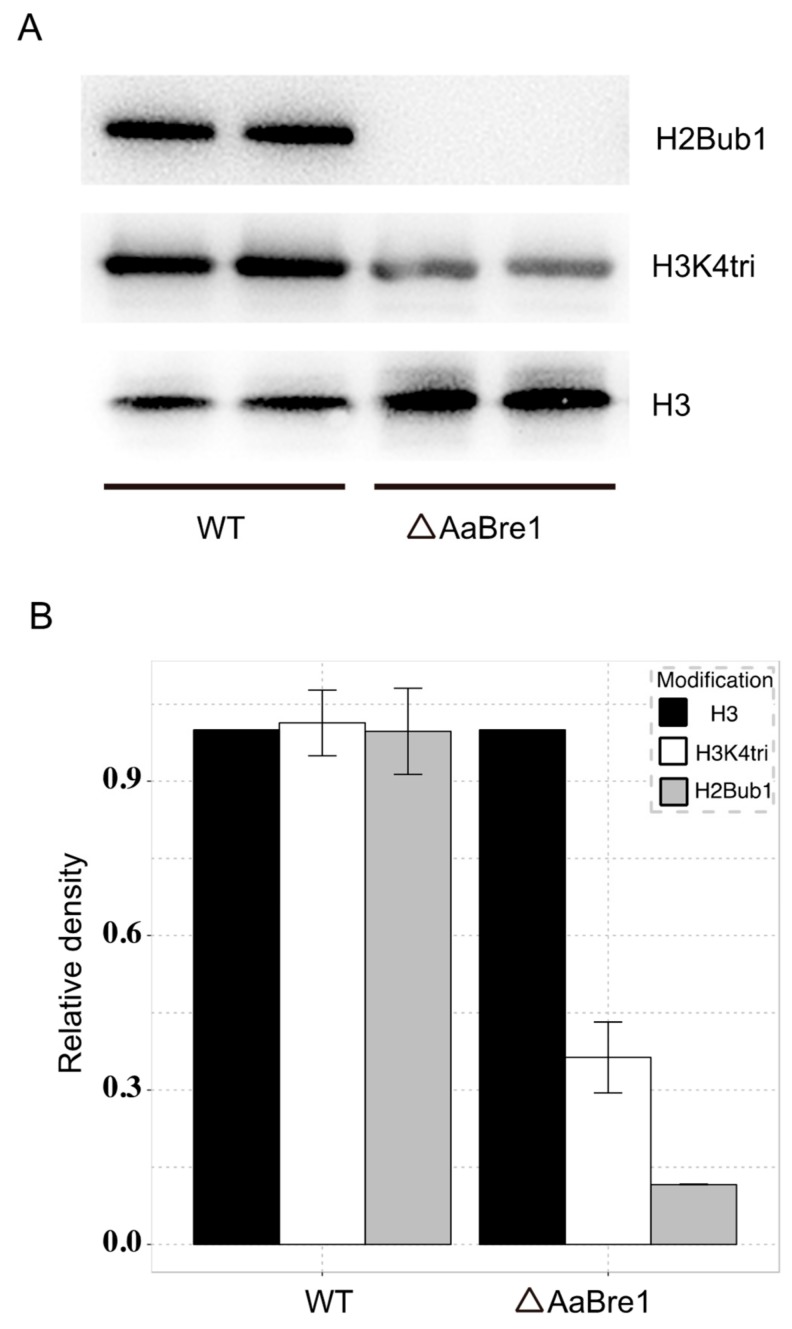
AaBre1 regulates H2B monoubiquitination and H3K4 trimethylation. (**A**) Western blot analyses demonstrate the absence in the ΔAaBre1 strain of H2B monoubiquitination and a reduced level of H3K4 trimethylation. (**B**) The relative intensity of H2B monoubiquitination and H3K4 trimethylation were analyzed by normalizing the amount of H2B monoubiquitination and H3K4 trimethylation with that of histone 3 using Tanon Image software. Results are shown as mean+SE from the two independent samples. Histone 3 (H3); H3K4 trimethylation (H3K4tri); H2B monoubiquitination (H2Bub1).

**Figure 6 genes-11-00229-f006:**
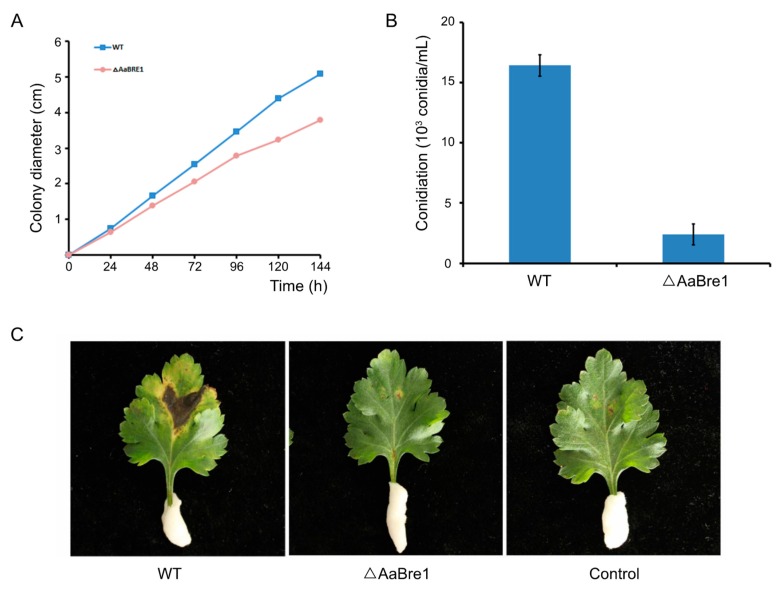
H2B monoubiquitination regulates mycelium growth, sporulation and pathogenesis in *A. alternata*. (**A**) The diameter of mycelial mats developed over time by the WT and ΔAaBre1 strains. (**B**) Conidial numbers produced by the WT and ΔAaBre1 strains grown on potato dextrose agar (PDA) for seven days. Whiskers indicate the SE (*n =* 3). (**C**) *Chrysanthemum* leaves inoculated with either the WT or the ΔAaBre1 strain. Control: mock inoculation with water.

**Figure 7 genes-11-00229-f007:**
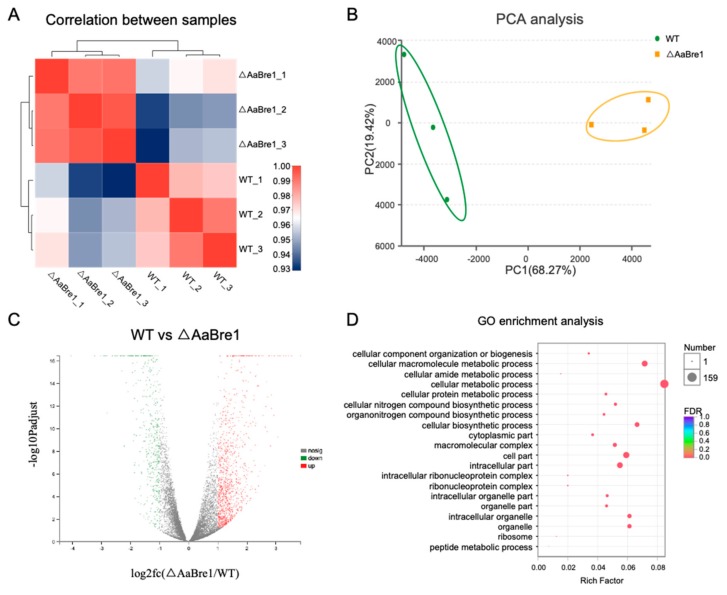
RNA sequencing analysis of the transcriptomes between WT and ΔAaBre1 strains. (**A**) Correlation analysis between samples. (**B**) PCA analysis between samples. (**C**) Volcano plot. Each strain has three replicates. (**D**) GO enrichment analysis of the differential expression genes (DEGs).

**Figure 8 genes-11-00229-f008:**
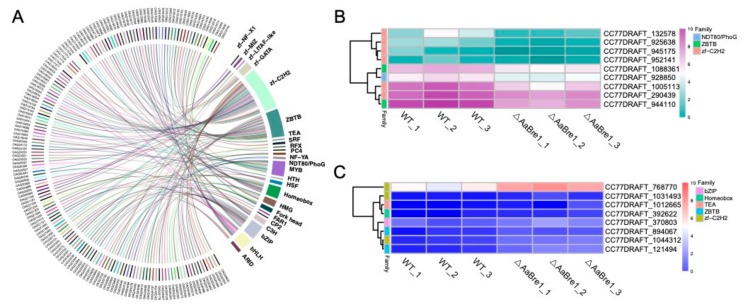
Identification of the differentially expressed transcription factors. (**A**) Classification of the transcription factors families represented in the *A. alternata* genome. (**B**,**C**) Heatmaps showing the contrasting behavior of (**B**) nine DEGs down-regulated in the mutant strain and (**C**) eight DEGs up-regulated in the mutant strain.

**Figure 9 genes-11-00229-f009:**
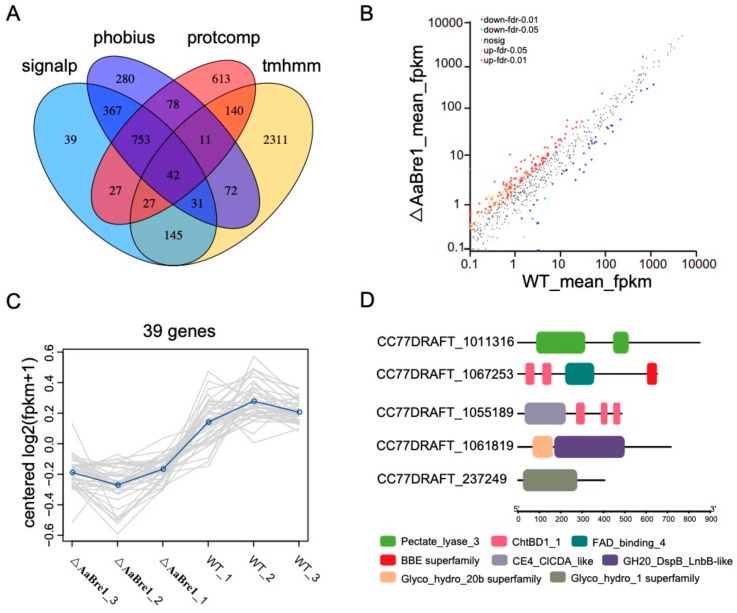
Expression analysis of secreted protein effectors between WT and ΔAaBre1. (**A**) Identification of the secreted proteins effectors in *A. alternata* genome. (**B**) Scatter plot of different expressed secreted protein effectors between WT and ΔAaBre1. (**C**) Sub-cluster of down-regulated secreted protein effectors. (**D**) Domain analysis of the down-regulated secreted protein effectors.

**Figure 10 genes-11-00229-f010:**
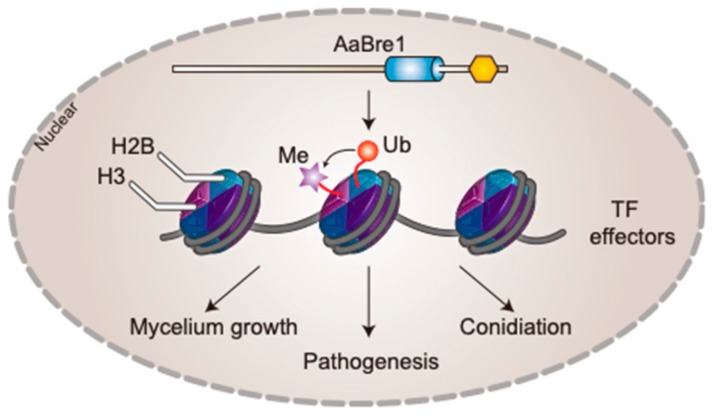
A proposed model of genetic networks regulated by AaBre1-mediated H2B monoubiquitination in *A. alternata*.

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
