# Peer review of "Ubiquitin E3 Ligase AaBre1 Responsible for H2B Monoubiquitination Is Involved in Hyphal Growth, Conidiation and Pathogenicity in Alternaria alternata"

_genes, 2020, doi:10.3390/genes11020229_

Round 1

Reviewer 1 Report

Dear authors,

I have read the manuscript entitled "Ubiquitin E3 ligase AaBre1 responsible for H2B monoubiquitination involves in hyphal growth, conidiation and pathogenicity in Alternaria alternata" with great interest". The authors have done quite some work in elucidating the role of AaBre1on hyphal growth, conidiation and pathogenicity of A. alternata. Similar epigenetic research in fungal pathogens is limited and thus this study is valuable. My only reservations are related with discussion, presentation of results and language quality of the manuscript. Text needs serious language editing in order to convey the correct meaning  and give justice to the great amount of experimentation done. In the title, the word "involves" should be "is involved". Introduction is too short and provides minimal backround regarding H2B epigenetic mechanism in context. Also,no aim of the study is presented  at the end of the introduction (where is typically placed). Discussion is awkward and often falls to pitfalls (e.g. at line 337-338: the authors generalize  the difference of H2B and H3K between A. alternata and plants taking into account only limited evidence). The last paragraph of the discussion  section is actually repeated in the conclusion section. Most section titles contain syntax or grammatical errors. Overall, I suggest that the authors get a language service to improve the quality of the manuscript. Concluding, I think that the manuscript needs major revision before being accepted in  the Genes journal.

Author Response

Point 1: I have read the manuscript entitled "Ubiquitin E3 ligase AaBre1 responsible for H2B monoubiquitination involves in hyphal growth, conidiation and pathogenicity in Alternaria alternata" with great interest". The authors have done quite some work in elucidating the role of AaBre1on hyphal growth, conidiation and pathogenicity of A. alternata. Similar epigenetic research in fungal pathogens is limited and thus this study is valuable. My only reservations are related with discussion, presentation of results and language quality of the manuscript. Text needs serious language editing in order to convey the correct meaning and give justice to the great amount of experimentation done. In the title, the word "involves" should be "is involved". Introduction is too short and provides minimal backround regarding H2B epigenetic mechanism in context. Also,no aim of the study is presented  at the end of the introduction (where is typically placed). Discussion is awkward and often falls to pitfalls (e.g. at line 337-338: the authors generalize the difference of H2B and H3K between A. alternata and plants taking into account only limited evidence). The last paragraph of the discussion section is actually repeated in the conclusion section. Most section titles contain syntax or grammatical errors.

 Response 1: Thanks a lot for your comments. Based on your suggestions, we modified the whole paper including the introduction, results, discussion and conclusion. The modification are as follows:

(1) In the title, the word “involves” was changed to “is involved”.

(2) We added more background regarding H2B epigenetic mechanism; please kindly check them at lines 41-49.

(3) The aim of the study is presented at the end paragraph of the introduction; please kindly check them at lines 77-81.

(4) We deleted the repeated conclusion section in the manuscript and modified the discussion section; please kindly check them at lines 431-511.

(5) We checked and modified the section titles; please kindly check them at lines 171, 198, 218, 310, 333 and 387.

Point 2: Overall, I suggest that the authors get a language service to improve the quality of the manuscript. Concluding, I think that the manuscript needs major revision before being accepted in the Genes journal.

Response 2: Thank you very much for your comments. Based on your suggestions, we got a a language service to improve the quality of the manuscript. All the changes have been highlighted in yellow in the paper.

Reviewer 2 Report

The authors identified the AaBRE1 gene, which encoded histone H2B monoubiquitination E3 ligase and found that AaBRE1 regulated hyphal growth, conidiation, and pathogenicity in Alternaria alternate. These authors provided several approaches to demonstrate their conclusion.

This manuscript needs to be edited for English. The data in this manuscript were clearly presented.  The subject of this report is quite interesting.

Author Response

Point 1:

The authors identified the AaBRE1 gene, which encoded histone H2B monoubiquitination E3 ligase and found that AaBRE1 regulated hyphal growth, conidiation, and pathogenicity in Alternaria alternate. These authors provided several approaches to demonstrate their conclusion.

This manuscript needs to be edited for English. The data in this manuscript were clearly presented.  The subject of this report is quite interesting.

Response 1: Thank you very much for your comments. Based on your suggestions, we checked and carefully edited the manuscript for English. All the changes have been highlighted in yellow in the paper.

Round 2

Reviewer 1 Report

The authors have improved the manuscript according to my suggestions and I now feel that it is acceptable for publication with minor modifications.

Specifically:

Although improved, the aim of the study in the introduction section still reports results instead of clearly stating the aim of the study in a concise way. Please revise. Line 359: insert "which was" between "factor" and "downregulation" Line 365: replace "contain" with "containing" Line 364: insert "present" between "In the" and " study" Line 366-367: replace "Among our identified DEGs, we also found two genes" with "Among the identified DEGs, two genes were  also found" Line 386: Replace "Our study opens distinctive pathways H2Bub1 played in regulating fungal growth" with "Our study opens distinctive pathways regarding the role that H2Bub1 plays in regulating fungal growth.."

Author Response

Point 1: The authors have improved the manuscript according to my suggestions and I now feel that it is acceptable for publication with minor modifications.

Specifically:

Although improved, the aim of the study in the introduction section still reports results instead of clearly stating the aim of the study in a concise way.

Response 1: Thank you very much for your comments. Based on your suggestions, we modified the last paragraph of the introduction, All the changes have been highlighted in yellow in the paper; please kindly check them at lines 63-65.

Point 2: Please revise. Line 359: insert "which was" between "factor" and "downregulation" Line 365: replace "contain" with "containing" Line 364: insert "present" between "In the" and " study" Line 366-367: replace "Among our identified DEGs, we also found two genes" with "Among the identified DEGs, two genes were  also found" Line 386: Replace "Our study opens distinctive pathways H2Bub1 played in regulating fungal growth" with "Our study opens distinctive pathways regarding the role that H2Bub1 plays in regulating fungal growth.." 

Response 2: Thank you very much for your comments. Based on your suggestions, we modified the manuscript. All the changes have been highlighted in yellow in the paper. The modification are as follows:

(1) "which was" was inserted between "factor" and "downregulation"; please kindly check them at line 354.

(2) "contain" was replaced with "containing"; please kindly check them at line 358.

(3) "present" was inserted between "In the" and " study"; please kindly check them at line 359.

(4) "Among our identified DEGs, we also found two genes" was replaces with "Among the identified DEGs, two genes were also found"; please kindly check them at lines 361-362.

(5) "Our study opens distinctive pathways H2Bub1 played in regulating fungal growth" was replaced with "Our study opens distinctive pathways regarding the role that H2Bub1 plays in regulating fungal growth."; please kindly check them at line 382.